# AutoPR: Automatically Pull Request Generation for Fix Issued Bugs of CodeBase

## Abstract

Over the past few decades, researchers have made significant strides in automating software development processes. This evolution has transformed the way software is created, maintained, and enhanced. Recently, the integration of Large Language Models (LLMs) into software development has opened new horizons. Researchers have investigated the potential of LLMs and demonstrated that they provide strong performance gains. These models can understand natural language instructions, generate code snippets, and even identify and fix bugs, thereby streamlining the development process. However, software engineering encompasses more than just coding; it involves the continuous improvement of programs to facilitate software maintenance and evolution. This includes tasks like program repair to fix bugs and feature additions to enhance functionality. Traditional automation tools often fall short in these areas, highlighting the need for more advanced solutions. Inspired by these insights, we have developed a novel automated program repair method called AutoPR. AutoPR represents a new generation of AI software engineers, leveraging routing algorithms, in-memory caching, and collaborative agent technologies. Its design addresses the current efficiency bottlenecks and quality issues faced in software development.

## 1 Introduction

Automating various tasks within software engineering has long been a goal for both researchers and practitioners (11; 30). Over the years, significant advances have been made in areas such as automated test generation, program repair, and even automatic code generation through large language models (LLMs) (2; 3). However, despite these developments, fully autonomous software engineering—where an AI system independently manages the entire lifecycle of a software project—remains an unsolved challenge. One of the main hurdles lies in dealing with ambiguous natural language requirements and seamlessly integrating generated code into complex, large-scale projects (20; 22).

Previous methods have made strides by leveraging LLMs and tools like GitHub Copilot, which provide code suggestions and assist with simple programming tasks (13). However, these approaches often struggle with deeper integration issues, such as code refactoring, comprehensive bug fixing, and code optimization. The trustworthiness of automatically generated code remains a concern, especially when such code must fit within large-scale, ongoing projects (23). These tools have yet to fully address the challenges of scaling across entire codebases, handling performance improvements, and autonomously generating new code frameworks for upcoming projects. The remaining challenges in realizing fully autonomous software engineering include effectively understanding large-scale codebases, automating complex refactoring and optimization tasks, intelligently detecting and fixing bugs, and generating project structures from scratch based on natural language requirements (8; 4). Moreover, the ability to continuously learn and adapt as the software evolves is still lacking in existing tools.

To illustrate the limitations of current approaches, we present a motivating example using our proposed tool, **AutoPR**. Figure 2 demonstrates the workflow of AutoPR on a feature addition task from the Django issue tracker, classified as a "New feature" in the tracker and included in SWE-bench lite with the ID "django-13933." This issue requests adding support to the `ModelChoiceField` class so that it "displays the value of the invalid choice when raising a validation error." AutoPR operates in two main stages: context retrieval and patch generation. In the

context retrieval stage, AutoPR identifies related classes and methods (e.g., `ModelChoiceField`, `ModelMultipleChoiceField`) and retrieves relevant code segments using APIs to search the Abstract Syntax Tree (AST) of the project. After gathering sufficient context, AutoPR generates a patch that modifies the `to_python` method to include the invalid value in the error message. The generated patch is validated and optimized using an automated refactoring tool, ensuring compatibility and adherence to project standards. This example demonstrates how AutoPR addresses complex feature addition tasks, something existing LLM-based tools often struggle with.

Our approach is guided by the intuition that combining LLMs with advanced algorithms such as routing algorithms, in-memory caching, and collaborative agent technologies can address these gaps. By augmenting the capabilities of LLMs with specialized techniques, we can enable more effective code understanding, optimization, and problem-solving across large projects. The motivation behind this work is to alleviate the manual, time-consuming aspects of software development, allowing developers to focus on higher-level tasks while the system handles routine maintenance, bug fixing, and performance improvements.

We introduce **AUTOPR**, a tool designed to automate complex software engineering tasks such as code refactoring, optimization, bug localization, and zero-shot code generation. AUTOPR enhances LLMs with advanced program representations like Abstract Syntax Trees (ASTs) and call graphs to deeply analyze and optimize codebases. The use of routing algorithms and in-memory caching allows for efficient navigation through large-scale projects, while collaborative agent technologies enable AUTOPR to perform multiple tasks simultaneously, such as analyzing code, refactoring, and fixing bugs (32).

To evaluate AUTOPR, we applied it to various large-scale codebases and new project requirements. Our tool demonstrated significant improvements in development efficiency, with faster refactoring times, more accurate bug fixes, and high-quality zero-shot code generation. We conducted thorough evaluations by measuring the correctness of the generated code, runtime performance improvements, and developer feedback in real-world projects.

Our contributions are as follows:

- We present a novel tool, AUTOPR, that enhances LLMs with advanced algorithms to autonomously handle complex software engineering tasks.

- We demonstrate how AUTOPR improves the efficiency of large-scale code analysis, refactoring, optimization, and bug fixing.

- We show that AUTOPR enables zero-shot code generation for new projects, significantly reducing development time and effort.

- Our approach effectively combines routing algorithms, in-memory caching, and collaborative agent technologies to manage large codebases and concurrent development tasks.

- We provide a comprehensive evaluation of AUTOPR 's performance, showing improvements in both code quality and developer productivity.

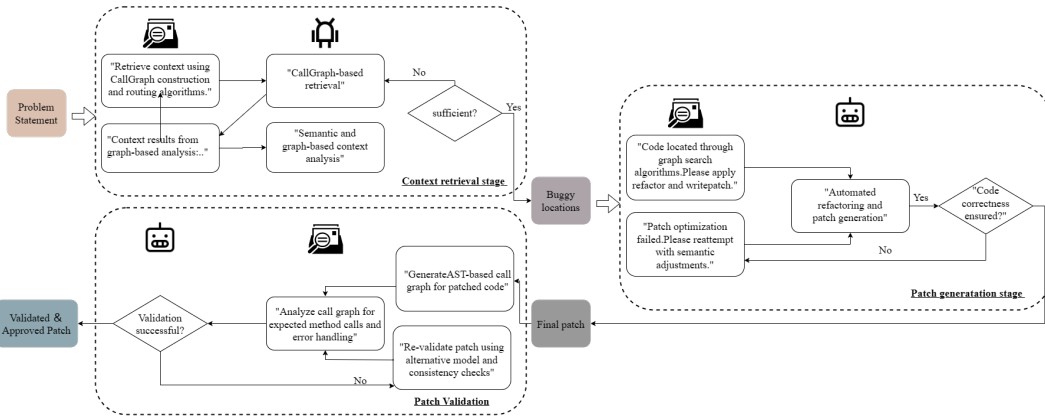

**Figure 1:** Overall workflow of AutoPR

## 2 METHODOLOGY

In this section, we present a novel approach for constructing and optimizing call graphs in Python projects by leveraging advanced mathematical optimization techniques. The methodology is designed to handle large-scale codebases with thousands of functions, classes, and files, ensuring efficient processing and accurate capture of code dependencies. The optimization framework not only minimizes computational complexity but also ensures scalability, making it applicable to projects of varying sizes and complexities.

### 2.1 CALL GRAPH CONSTRUCTION

A call graph is a directed graph $G = (V, E)$, where $V$ is the set of nodes representing entities such as files, functions, and classes in the codebase, and $E$ represents the directed edges that denote function calls or object instantiations between these components. The construction of this graph is crucial for visualizing and analyzing the structure of the software, as it captures the flow of execution across different parts of the code. The primary objective is to accurately capture all the relationships within a codebase while minimizing the error introduced during graph construction, ensuring that important connections are not overlooked.

To solve this optimization problem, we can introduce more complex reasoning and additional constraints to model the quantization process more precisely and consider the regularization of the low-rank matrices.

First, we explicitly specify the limitations of the quantized matrix $Q$, that is, its elements can only take discrete quantization level values. Let the set of quantization levels be $\mathcal{Q} = \{q_1, q_2, \ldots, q_N\}$. Therefore, we can reformulate the optimization problem as:

$$\min_{Q,A,B} \quad \|W - Q - AB^\top\|_F^2 + \lambda_A\|A\|_F^2 + \lambda_B\|B\|_F^2, \tag{1}$$

$$\text{s.t.} \quad Q_{ij} \in \mathcal{Q}, \quad \forall i,j, \quad \text{rank}(AB^\top) = r. \tag{2}$$

Here, $\lambda_A$ and $\lambda_B$ are regularization parameters controlling matrices $A$ and $B$, which help prevent overfitting and ensure the stability of the low-rank matrices.

To further complicate the problem, we can model the quantization process as an optimization variable. Suppose we introduce indicator variables $Z_{ij}^k$, satisfying:

$$Q_{ij} = \sum_{k=1}^{N} q_k Z_{ij}^k, \tag{3}$$

where $Z_{ij}^k \in \{0, 1\}$ and $\sum_{k=1}^{N} Z_{ij}^k = 1$ holds for all $i, j$. In this way, our optimization problem becomes:

$$\min_{Z,A,B} \quad \left\|W - \sum_{k=1}^{N} q_k Z^{(k)} - AB^\top\right\|_F^2 + \lambda_A\|A\|_F^2 + \lambda_B\|B\|_F^2,$$

$$\text{s.t.} \quad Z_{ij}^k \in \{0, 1\}, \quad \sum_{k=1}^{N} Z_{ij}^k = 1, \quad \forall i, j, \tag{4}$$

$$\text{rank}(AB^\top) = r.$$

Here, $Z^{(k)}$ is the matrix composed of $Z_{ij}^k$. By incorporating the quantization process into the optimization framework, we make the problem more complex and precise.

Additionally, we can consider introducing sparsity constraints to further reduce storage and computational overhead. By adding $\ell_1$ regularization terms to the objective function, we get:

$$\min_{Z,A,B} \quad \left\| W - \sum_{k=1}^{N} q_k Z^{(k)} - AB^\top \right\|_F^2 + \lambda_1 \left( \|A\|_1 + \|B\|_1 \right) + \lambda_2 \left( \|A\|_F^2 + \|B\|_F^2 \right),$$

$$\text{s.t.} \quad Z_{ij}^k \in \{0,1\}, \quad \sum_{k=1}^{N} Z_{ij}^k = 1, \quad \forall i,j. \tag{5}$$

Here, $\lambda_1$ and $\lambda_2$ are regularization parameters, and $\| \cdot \|_1$ denotes the sum of the absolute values of the matrix elements, encouraging sparse solutions.

Finally, to handle the non-convexity and non-differentiability of the quantization function, we can use approximate continuous functions (such as soft quantization) to replace hard quantization, allowing the use of gradient-based optimization methods to solve it.

## 2.2 Enhanced Optimization Procedure

To solve this optimization problem, we adopt an advanced alternating optimization strategy that iterates between updating the quantized matrix $Q$ and refining the low-rank matrices $A$ and $B$, incorporating complex constraints and regularization techniques to enhance the solution quality.

The optimization procedure is divided into two key steps:

**Step 1: Quantization Update with Constraints**

At each iteration $t$, we update the quantized matrix $Q_t$ by solving a constrained optimization problem that explicitly models the quantization process. We define the set of quantization levels $\mathcal{Q} = \{q_1, q_2, \ldots, q_N\}$ and introduce binary indicator variables $Z_{ij}^k$, where $Z_{ij}^k \in \{0,1\}$ and $\sum_{k=1}^{N} Z_{ij}^k = 1$ for all $i,j$. The quantized matrix $Q_t$ is expressed as:

$$Q_{t,ij} = \sum_{k=1}^{N} q_k Z_{ij}^k, \tag{6}$$

To update $Q_t$, we solve the following integer programming problem:

$$\min_{Z} \quad \left\| W - A_{t-1} B_{t-1}^\top - \sum_{k=1}^{N} q_k Z^{(k)} \right\|_F^2,$$

$$\text{s.t.} \quad Z_{ij}^k \in \{0,1\}, \quad \sum_{k=1}^{N} Z_{ij}^k = 1, \quad \forall i,j. \tag{7}$$

This problem is combinatorial in nature and NP-hard due to the binary constraints. To make it tractable for large-scale problems, we can employ approximation algorithms such as:

- **Relaxation to Continuous Variables**: Relax $Z_{ij}^k$ to be in $[0,1]$, turning the problem into a convex optimization problem. After solving, apply thresholding to obtain binary values. - **Greedy Algorithms**: Update $Z_{ij}^k$ element-wise by selecting the quantization level that minimizes the local reconstruction error. - **Alternate Direction Method of Multipliers (ADMM)**: Decompose the problem into subproblems that are easier to solve iteratively.

**Step 2: Low-Rank Approximation with Regularization and Sparsity Constraints**

After obtaining $Q_t$, we compute the residual matrix: $R_t = W - Q_t$.

We then seek to find low-rank matrices $A_t$ and $B_t$ that approximate $R_t$ while incorporating regularization and sparsity constraints to prevent overfitting and enhance interpretability. We solve the following optimization problem:

$$\min_{A_t, B_t} \quad \left\| R_t - A_t B_t^\top \right\|_F^2 + \lambda_A \|A_t\|_F^2 + \lambda_B \|B_t\|_F^2 + \lambda_1 \left( \|A_t\|_1 + \|B_t\|_1 \right),$$

$$\text{s.t.} \quad \text{rank}(A_t B_t^\top) \leq r. \tag{8}$$

The $\ell_1$ regularization terms $\lambda_1 \left( \|A_t\|_1 + \|B_t\|_1 \right)$ encourage sparsity in the factors $A_t$ and $B_t$, which can be crucial in high-dimensional settings to improve model interpretability and reduce overfitting.

To solve this non-convex optimization problem, we can employ iterative algorithms such as:

- **Alternating Minimization**: Fix $B_t$ and solve for $A_t$, then fix $A_t$ and solve for $B_t$, iteratively updating each while keeping the other constant. - **Proximal Gradient Methods**: Incorporate proximal operators to handle the non-smooth $\ell_1$ regularization terms, updating $A_t$ and $B_t$ simultaneously.

The update rules for $A_t$ and $B_t$ in proximal gradient methods can be expressed as:

$$A_t \leftarrow \text{prox}_{\alpha\lambda_1 \|\cdot\|_1} \left( A_t - \alpha \nabla_{A_t} f(A_t, B_t) \right), \tag{9}$$

$$B_t \leftarrow \text{prox}_{\alpha\lambda_1 \|\cdot\|_1} \left( B_t - \alpha \nabla_{B_t} f(A_t, B_t) \right), \tag{10}$$

where $\alpha$ is the step size, $f(A_t, B_t) = \left\| R_t - A_t B_t^\top \right\|_F^2 + \lambda_A \|A_t\|_F^2 + \lambda_B \|B_t\|_F^2$, and $\text{prox}_{\gamma\|\cdot\|_1}(\cdot)$ denotes the proximal operator for the $\ell_1$ norm.

The gradients are computed as:

$$\nabla_{A_t} f = -2(R_t - A_t B_t^\top) B_t + 2\lambda_A A_t, \tag{11}$$

$$\nabla_{B_t} f = -2(R_t - A_t B_t^\top)^\top A_t + 2\lambda_B B_t. \tag{12}$$

The proximal operator for the $\ell_1$ norm is defined component-wise as:

$$\text{prox}_{\gamma\|\cdot\|_1}(x_i) = \text{sign}(x_i) \cdot \max(|x_i| - \gamma, 0), \tag{13}$$

which performs soft-thresholding to promote sparsity.

The overall optimization algorithm is as shown in appendix.

## 2.3 INTEGRATION INTO CALL GRAPHS

Once the matrices $Q_t$, $A_t$, and $B_t$ are updated, the optimized structure is integrated into the call graph. In this graph, each node represents a component in the code, such as a function, class, or file, while the edges represent the relationships between these components as determined by the non-zero entries in the final matrix $W$. The resulting call graph $G = (V, E)$ effectively captures function calls, object instantiations, and other execution flow dependencies within the codebase.

This optimized call graph provides an efficient and scalable representation of large-scale projects, making it a powerful tool for code analysis and refactoring tasks. By combining quantization with low-rank approximation, we are able to reduce the memory and computational requirements of handling complex codebases. Additionally, this process makes it easier to identify important dependencies and potential performance bottlenecks, facilitating optimizations and improvements in software projects.

The iterative nature of the optimization ensures that large-scale projects with thousands of nodes and edges can be handled without sacrificing performance. As the optimization refines the approximation, the graph becomes a more accurate reflection of the code structure, making it ideal for various software engineering tasks such as bug detection, refactoring, and dependency analysis.

## 3 EXPERIMENTAL SETUP

To evaluate the capabilities of **AutoPR** in resolving real-world software issues, we aim to answer the following research questions.

**RQ1**: To what extent can AutoPR automate software issue resolution like human developers?

**RQ2**: Can existing debugging or analysis techniques assist AutoPR?

**RQ3**: What are the challenges for AutoPR and fully automated program improvement in the future?

**Benchmark.** We evaluate AutoPR using the SWE-bench benchmark (10), which comprises a collection of real-life GitHub issues. The input for each instance includes the natural language description from the original GitHub issue and its corresponding buggy codebase. Details of SWE-bench are provided in Section 2.2.

**Baselines and Evaluation Metrics.** We selected two LLM-based agent systems, Devin and Swe-agent (32), as baselines to compare their performance against AutoPR. Since we do not have access to Devin, we reference the most relevant reported results from their technical report. Swe-agent is publicly available as a GitHub repository, and we replicated it with default settings based on the provided scripts. To evaluate the effectiveness of the tools, we use (1) the percentage of resolved instances, (2) average time cost, and (3) average token cost. These evaluation metrics represent overall effectiveness, time efficiency, and economic efficacy in resolving real-world GitHub issues (10).

**Implementation and Parameters.** We use the state-of-the-art OpenAI GPT-4 (`gpt-4-0125-preview`) as the foundational inference model for AutoPR. The GPT-4 model is responsible for selecting search APIs to retrieve codebase context, refining the issue description, and writing a final patch. For the GPT-4 parameters, we set a low `temperature=0.2` and `max_tokens=1024` to produce relatively deterministic results and allow sufficient reasoning length for AutoPR; all other parameters remain at their default settings. Note that AutoPR does not have a time limit and terminates either when a patch is generated or when the cost of resolving an issue reaches two USD (4).

**System Environment.** All experiments are conducted on an x86_64 Linux server with Ubuntu 20.04 installed. graphicx

## 4 EXPERIMENT RESULTS

### 4.1 RQ1: OVERALL EFFECTIVENESS ON SWE-BENCH

We first measure the overall effectiveness of **AutoPR** and baselines with the number of resolved task instances in SWE-bench. Aiming to understand the extent to which current AI systems can automatically resolve real-life software issues, we provided only the natural language issue description and a local code repository checked out at the erroneous version as inputs. We repeated AutoPR's experiment three times and presented the average and total number of resolved software issues across the three runs. The average and total results are denoted as **AutoPR-avg** and **AutoPR-all** respectively (for brevity, we use **AutoPR** to denote AutoPR in this section). When reporting time and token/cost for AutoPR-all, we report the time and cost required for running each task three times. Since Devin was evaluated on a random 25% subset of SWE-bench (10), we also report results of AutoPR on this subset (referred to as "SWE-bench Devin subset"). Table 1 shows the overall result in full SWE-bench, SWE-bench Devin subset, and SWE-bench lite respectively. Figure 6 provides a visual summary of AutoPR's comparison with Swe-agent and Devin.

In the full SWE-bench, AutoPR-all (union from the three AutoPR runs) resolved **18.16%** of task instances, taking **1377 seconds** per task ( 23 minutes). In comparison, Swe-agent resolved **12.29%** of tasks in full SWE-bench, according to their report (32) (we did not replicate Swe-agent-rep on the full SWE-bench due to high cost).

The state-of-the-art closed-source baseline tool Devin (10) is evaluated on a random 25% subset of SWE-bench. To compare AutoPR with Devin, we report AutoPR's results on the 570 task instances Devin was evaluated on, taken from AutoPR's runs on full SWE-bench. In the SWE-bench Devin subset, the union of three runs of AutoPR successfully resolved **17.98%** of the task instances, which

is higher than Devin's **13.86%**. Figure 7a provides a more detailed exposition of the resolved tasks. Moreover, the times taken by AutoPR and Devin are comparable.

We performed another round of experiments with AutoPR on SWE-bench lite (300 instances). The results reported in Table 1 indicate that on average AutoPR-avg can resolve **18.35%** of task instances in SWE-bench lite, which is higher than the reported results from Swe-agent (**17.00%**). We also investigated the union of all resolved tasks in the three runs, in which the percentage of resolved task instances increased to **25.43%**.

**Table 1:** Results comparing Swe-agent, Devin, and AutoPR on various SWE-bench datasets

| Tools | Resolved Tasks | Avg Time | Avg Tokens |
|---|---|---|---|
| \multicolumn{4}{c}{Reported result on full SWE-bench (size=2294)} |
| **Swe-agent (32)** | **12.29% (282)** | 93 | - |
| **A-PR-avg** | 12.06% (275) | 460 | 78926 ($0.912) |
| **A-PR-all** | 18.16% (415) | 1377 | 236777 ($2.735) |
| \multicolumn{4}{c}{Reported result on SWE-bench Devin subset (size=570)} |
| **Devin (10)** | **13.86% (79)** | >600 | - |
| **A-PR-avg** | 11.92% (67) | 454 | 76748 ($0.888) |
| **A-PR-all** | 17.98% (102) | 1359 | 230243 ($2.666) |
| \multicolumn{4}{c}{Reported result on SWE-bench lite (size=300)} |
| **Swe-agent (32)** | **17.00% (51)** | 93 | 69976 ($0.739) |
| **A-PR-avg** | 18.35% (54) | 340 | 58682 ($0.4112) |
| **A-PR-all** | 25.43% (76) | 1022 | 221603 ($2.562) |

**Table 2:** Replication result on SWE-bench lite (size=300)

|  | In our environment | | In `Swe-agent` Docker | |
|---|---|---|---|---|
|  | **A-PR** | **Agent-Rep** | **A-PR** | **Agent-Rep** |
| **Run 1** | 18.22% (54) | 9.33% (28) | 10.00% (30) | 6.67% (20) |
| **Run 2** | 17.85% (53) | 11.00% (33) | 10.33% (31) | 7.00% (21) |
| **Run 3** | 18.98% (56) | 9.33% (28) | 10.67% (32) | 6.00% (18) |
| **All** | **25.43% (76)** | 14.67% (44) | 14.00% (42) | 9.00% (27) |

Since Swe-agent is publicly available, we also attempted to run Swe-agent on SWE-bench lite. We replicated Swe-agent with two USD as cost budget for conversation with LLM per task instance in our environment (denoted as Swe-agent-rep, it terminates either a patch is generated or reaches the two USD budget). Table 2 shows that when considering the union of all resolved tasks across three repetitions, AutoPR resolved **25.43%** out of the 300 tasks, whereas Swe-agent-rep resolved **14.67%**. We further analyzed the commonly and uniquely resolved instances between AutoPR and Swe-agent-rep in Figure 7b, finding that AutoPR and Swe-agent-rep complement each other in different scenarios. AutoPR uniquely resolved **32** task instances, benefiting from the fine-grained code context search at the AST level to precisely locate the bug locations. Conversely, the main reason that AutoPR failed on the **9** unique instances resolved by Swe-agent-rep is unimplemented search APIs (e.g., search_file invoked in django-12286). In such cases, AutoPR can generate invalid search results when unimplemented APIs are invoked, implying more robust search APIs are desired for future improvement.

We are also interested in assessing the feasibility of deploying AutoPR in the real world in terms of time and economic cost. On average, AutoPR takes **460 seconds** and **78926 tokens** (equivalent to **0.912 USD**) to resolve one task instance in SWE-bench. In comparison, our replication experiments with Swe-agent-rep cost **69976 tokens** (equivalent to **0.739 USD**) per task instance. When considering the combined three repetitions, AutoPR takes **1377 sec-**

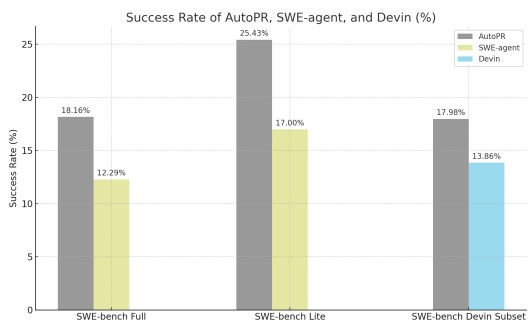

**Figure 2:** Summary of Results AUTOPR

**onds** ( 23 minutes) per task, which is below
the 30-60 minute time limit deemed acceptable
by developers for automated repair tools (31).
Looking into the **76** issues resolved by AutoPR
in SWE-bench lite, it costs on average 2.77 days for developers to create pull requests for **57** issues,
and the other **10** issues take even longer to be closed by developers (ranging from 34 - 4023 days).
The short response time and low cost show the significant potential for AutoPR to act as a first step
in future autonomous bug fixing.

**Table 3:** Result of A-PR-val, A-PR-val-callgraph on SWE-bench lite, in one run only.

| Tools | Resolved Tasks | Avg Time | Avg Tokens |
|---|---|---|---|
| **A-PR-avg** | 18.35% (55) | 340 | 58682 ($0.4112) |
| **A-PR-val** | 19.36% (58) | 459 | 91600 ($1.057) |
| **A-PR-val-callgraph** | 23.15% (69) | 491 | 79649 ($0.923) |

Overfitting is a well-known challenge in the Automated Program Repair community (28). A pro-
gram patch that passes the given test suite is said to be plausible. However, a plausible patch is
deemed as overfitting if it fails to conform to the developer's intent. Otherwise, it is deemed as
correct. To further understand the patch quality of AutoPR and baselines, we manually verify the
correctness of task-resolving (i.e., plausible) patches in SWE-bench lite. Since three repetitions are
performed, we consider a task to have a correct patch if any of the three repetitions produced a
correct patch. A plausible patch is correct if it is semantically equivalent to the developer patch.
In this verification process, at least two authors of the paper cross-validated each patch, and any
disagreement was resolved with another author. Overall, on SWE-bench lite, AutoPR has a correct-
ness rate of **65.7%** (44 correct/67 plausible). Swe-agent-rep has a slightly higher correctness rate of
**72.7%** (32/44), but the absolute number of correctly resolved tasks is smaller than AutoPR. Finally,
the correctness rate of Devin on SWE-bench Devin subset is **53.2%** (42/79). We observed that the
vast majority of AutoPR's overfitting patches (all but 2 of the overfitting patches) modify the same
methods as the developer patches, but the code modifications are incorrect. This means that even
the overfitting patches from AutoPR are useful to the developer, since it helps in localization. The
main causes of wrong modifications are the limits of the LLM's capability or insufficient context.

—

## 4.2 RQ2: EFFECT OF CALLGRAPH

In this research question, we aim to understand whether program analysis techniques such as **Call-
Graph** can benefit the workflow of **AutoPR**. Different from RQ1, here we simulate a common
scenario in program repair where AutoPR has access to the complete test-suite of the target task
instance. We use the developer-written test cases for each task instance (provided in SWE-bench
lite) as the test-suite. To assess the effect of **CallGraph**, we conduct two sets of experiments:

1. **AutoPR** uses the test-suite for patch validation during the patch generation retry-loop.
2. In addition to the first setup, we provide **CallGraph** results (top-5 relevant methods based
   on call relationships) to AutoPR at the beginning of the context retrieval stage.

We denote these two settings as **AutoPR-val** and **AutoPR-val-callgraph**. The patch validation
works as follows: when a patch is generated by the LLM agent, the test-suite is executed on the
patched program. If the patch fails to pass the complete test-suite, AutoPR re-invokes the patch
generation agent to write a new patch. This validation loop is configured to run at most three times.

**Results:** Table 3 shows that, with the additional information provided by **CallGraph**, the number of
resolved tasks increased from **58** to **69** (i.e., from **19.36%** to **23.15%** resolved rate on SWE-bench
lite). Compared to the validation-only setting, adding the **CallGraph** component helps to resolve **11**
additional unique task instances. Moreover, when comparing with task instances resolved in all other
runs combined (i.e., **AutoPR-all** and **AutoPR-val**) on SWE-bench lite, **AutoPR-val-callgraph** still
provides valuable insights by highlighting key method interactions that are not obvious from the
issue description alone.

For example, when call relationships between methods are revealed through the **CallGraph** analysis, AutoPR is able to exploit these relationships and improve context retrieval, helping it to generate more accurate patches. This suggests that a structure-based analysis such as **CallGraph** can complement the agent's workflow by revealing deeper inter-method dependencies that may not be fully detailed in the issue description.

## 4.3 RQ3: CHALLENGES ON REAL-LIFE TASKS

In this research question, we analyze the task instances in SWE-bench lite that **AutoPR** failed to resolve and provide a taxonomy of the issue characteristics to highlight the practical challenges in achieving fully automated software improvement. Our taxonomy consists of challenges in the fault localization stage and patch generation stage. Specifically, for each task, we analyze the best run in the three repetitions and classify each of the 300 tasks into one of the following:

- **Success**: The generated patch resolves the issue.

- **Wrong patch**: The generated patch modifies all methods that are modified in the developer patch. This means the patch content is wrong, but the patch location(s) are correct.

- **Wrong location in correct file**: The generated patch modifies the correct file but wrong location(s) in the file.

- **Wrong file**: The generated patch modifies the wrong file.

- **No patch**: No patch is generated from the retrieved context.

Figure 9 shows the distribution of the 300 tasks in SWE-bench lite. **AutoPR** resolves **25.43%** of the issues ("Success"), as mentioned in Section 1. The fail-to-resolve cases are included in the remaining four categories. In **31.7%** of the tasks, **AutoPR** correctly identified all patch locations (at the method level), but did not produce a correct patch ("Wrong patch"). More fine-grained intra-procedural analysis and specification inference techniques can play a significant role in improving these cases by providing the patch generation agent with more method-level repair guidance.

In the other three categories, the fault localization could not pinpoint all the locations to be modified. In **20.7%** of the tasks, a patch is generated in the correct file but at wrong methods/classes in the file ("Wrong location in correct file"). In some of these runs, the developer patch modifies multiple methods, but the generated patch did not modify all of them.

In the other categories, a patch could not be generated in the correct file—in **18.7%** of the tasks, a patch is generated in wrong files, and in **6.7%** of the tasks, there is no applicable patch ("Wrong file" and "No patch"). We manually inspected some tasks in these two categories and observed that their issue description mentions few methods/classes/files in the codebase. Instead, some of them contain short examples to reproduce the issue. For these tasks, one possibility is to generate a comprehensive test-suite based on the issue description and then use execution information from the test-suite (e.g., **CallGraph**) to reveal suspicious program locations. On the other hand, some other tasks do not contain reproducible examples and only consist of natural language descriptions. For these tasks, some human involvement might be helpful. Developers could focus on these tasks.

## 5 RELATED WORK

### 5.1 PROGRAM REPAIR TECHNIQUES

Test-suite-based Automated Program Repair (APR) methods like GenProg use search-based techniques, while semantic-based repair approaches such as Angelix leverage program synthesis to generate patches by solving symbolic constraints (30; 21; 19). Other methods like SemFix utilize symbolic execution, and Prophet ranks patches using probabilistic models (14). Recent APR methods employ deep learning, such as CoCoNuT's neural machine translation framework and Tufano's sequence-to-sequence learning models (16; 29). However, they often rely on high-quality test suites, prompting the exploration of heuristic-based repairs using static analysis or search-based techniques for scenarios lacking comprehensive test coverage (6; 12).

## 5.2 LLMs for Program Repair

LLMs like Codex and GPT-3 have shown promise in APR by generating patches based on code context and natural language bug descriptions (3; 4). While these models excel in zero-shot learning, they require pre-identified buggy statements, posing challenges in large-scale projects lacking fault localization capabilities (25; 18). Additionally, issues such as buggy input detection, test case reliance, and limited semantic understanding remain challenges for LLMs in APR (33; 31).

## 5.3 SWE-bench Dataset

The SWE-bench dataset offers a more realistic evaluation of LLMs in software engineering by including GitHub issues and corresponding pull requests from large projects like Django and SymPy (9; 24). It focuses on complex tasks such as understanding, refactoring, and bug fixing in mature codebases, setting it apart from benchmarks like HumanEval and CodeXGLUE (7; 15). SWE-bench lite enables faster evaluation while maintaining the challenge of real-world software tasks.

## 5.4 Advanced Techniques in Code Analysis and Refactoring

AutoPR leverages ASTs, call graphs, and graph-based algorithms to navigate large codebases and perform automated refactoring and optimization (26; 27). Techniques such as static and dynamic code analysis are used to identify performance bottlenecks (1; 5), while routing algorithms and in-memory caching enhance efficiency in large projects (27). Collaborative agent technologies enable concurrent handling of tasks like bug fixing and optimization, reducing developer workload and improving collaboration (31; 32).

## 6 Conclusion

As a new-generation AI software engineer, **AutoPR** integrates advanced technologies like graph analysis (26; 27), memory caching, semantic understanding, and multi-agent collaboration (32), excelling in tasks such as code understanding, generation, and optimization (3; 13). Unlike existing tools, AutoPR offers clear advantages in scalability, comprehensiveness, and collaboration, handling projects of any size. By leveraging graph-based analysis and memory caching, it efficiently processes large codebases, overcoming limitations faced by traditional tools (22; 28). AutoPR also conducts deep semantic analysis, allowing it to refactor code and optimize performance while ensuring correctness (2; 17). Its multi-agent system mirrors human teamwork, enabling parallel handling of tasks like bug fixing and feature implementation (32). This combination of scalability, accuracy, and collaboration positions AutoPR as a vital tool for developers, with the potential to greatly enhance productivity and code quality, driving the future of intelligent software engineering (11; 31).

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
