# OpenReview forum: "AutoPR: Automatically Pull Request Generation for Fix Issued Bugs of CodeBase"
_ICLR.cc/2025/Conference — ICLR 2025 Conference Withdrawn Submission_

### Official Review · Reviewer_eJQA · 2024-11-03

**Soundness:** 2
**Presentation:** 1
**Contribution:** 2
**Rating:** 3
**Confidence:** 4

**Summary:**

The paper proposes AutoPR - a tool for automatically generating pull requests based on issue description. AutoPR uses LLMs and call graph-based context retrieval to automatically generate patches. Authors run AutoPR on the SWE-bench dataset and compare it to Swe-agent and Devin showing outperformance with Auto-PR-all.

**Strengths:**

1. **LLMs + Call Graphs**: Combines LLMs with call graphs and ASTs to improve PR generation from problem description.
2. **Improved Task Resolution Rate**: Demonstrated improved task completion on the SWE-bench dataset compared to Devin and SWE-Agent using Auto-PR-all.

**Weaknesses:**

1. **Very Poor Presentation**: I list the details of the presentation issues in the Questions section and mention major ones in other points below.
2. **Call Graph Construction**: Most of section 2 seems to be misplaced in this paper. I don't believe authors claim novelty in call graph construction. Authors: Could you explain the relevance of the detailed call graph construction description in Section 2 to the main contributions of AutoPR? If there are novel aspects in this process, please highlight them explicitly.
3. **Missing Routing Algorithms and In-Memory Caching** - Authors claim using routing algorithms and in-memory caching but do not present these parts of the system. Authors: Could you provide details on how routing algorithms and in-memory caching are implemented in AutoPR, and how they contribute to its performance?
4. **Auto-PR-Avg Performance Subpar** - While Auto-PR-all outperforms competitors, Auto-PR-avg underperforms competitors. Can you explain why Auto-PR-all is compared to single-run metrics of competitors? It would be helpful to see a comparison of Auto-PR-avg with competitors' single-run results, or Auto-PR-all with multiple runs of competitors if available.

**Questions:**

Most of the questions refer to issues with presentation:
1. Figure 1 is unclear and not explained
2. “Figure 2 demonstrates the workflow of AutoPR on a feature addition task from the Django issue tracke” – it does not. Figure 2 is “Summary of Results”. Was Figure 2 that demonstrated workflow removed?
3. I don’t understand why sections 2.1 and 2.2 are in this paper. Is there something novel in call graph construction? If not, then why these sections are in the paper? If there is some novelty, what is it? Authors do not mention any novelty in call graph construction in their contributions.
4. There are further issues with presentation in sections 2.1 and 2.2, which I'll mention for posterity:
Can you explain what you mean by: “To solve this optimization problem, we can introduce more complex reasoning and additional con-
straints to model the quantization process more precisely and consider the regularization of the low-rank matrices”. Can you explain what you are trying to say with "To further complicate the problem”?
5. “Details of SWE-bench are provided in Section 2.2” – no they are not. Section 2.2 is “ENHANCED OPTIMIZATION PROCEDURE” and does not talk about SWE-bench. Did you mean section 5.3?
6. “(for brevity, we use AutoPR to denote AutoPR in this section” – did you mean A-PR?
7. Why do you have Figure 2? It repeats the same information as Table 1.
8. “This means that even the overfitting patches from AutoPR are useful to the developer, since it helps in localization” - this rather tenuous argument. Could you provide more concrete evidence or examples to support the claim that overfitting patches are useful for localization? How does this compare to the localization capabilities of other approaches?
9. Figure 9 – there is no Figure 9.

---

### Official Review · Reviewer_3Cj1 · 2024-11-03

**Soundness:** 2
**Presentation:** 2
**Contribution:** 2
**Rating:** 3
**Confidence:** 5

**Summary:**

The paper introduces AutoPR, an automated program repair tool aimed at generating pull requests to fix bugs in codebases. AutoPR utilizes routing algorithms, in-memory caching, and collaborative agent technologies to enhance the performance and efficiency of large-scale software maintenance tasks. The tool was evaluated on the SweBench dataset, showcasing its capabilities in automating complex software engineering tasks.

**Strengths:**

The application of AutoPR to the SWE-Bench dataset provides a relevant testbed for evaluating its performance in real-world scenarios.

**Weaknesses:**

1. Writing and Presentation: The paper is poorly written and has formatting issues, including LaTeX symbol errors. These detract from the readability and professionalism required for such submissions.

2. Lack of Novelty: The paper fails to establish a clear distinction between AutoPR and existing methods such as AutoCodeRover[1], RepoUnderstander[2], and CodeXgraph[3]. A detailed comparison is necessary to highlight the novel contributions of AutoPR. Consider suggesting a comparison table of key features or capabilities between AutoPR and these systems(AutoCodeRover, RepoUnderstander, CodeXgraph) to highlight the distinctions.

3. Evaluation Metrics: The comparison using pass@3 with SWE-agent is questionable as it may not provide a fair evaluation. Table 1 indicates that AutoPR's average performance is inferior to that of SWE-agent, suggesting the need for more rigorous benchmarking. Please use evaluation metrics such pass@1 to provide a more rigorous and fair comparison between AutoPR and SWE-agent.

```
[1] Zhang Y, Ruan H, Fan Z, et al. Autocoderover: Autonomous program improvement[C]//Proceedings of the 33rd ACM SIGSOFT International Symposium on Software Testing and Analysis. 2024: 1592-1604.
[2] Ma Y, Yang Q, Cao R, et al. How to Understand Whole Software Repository?[J]. arXiv preprint arXiv:2406.01422, 2024.
[3] Liu X, Lan B, Hu Z, et al. Codexgraph: Bridging large language models and code repositories via code graph databases[J]. arXiv preprint arXiv:2408.03910, 2024.
```

**Questions:**

1. Can the authors provide a more detailed comparison with existing methods, particularly in terms of architectural differences and performance benchmarks?

2. Could the authors elaborate on the choice of evaluation metrics and justify the fairness of the comparisons made?

3. What are the limitations of AutoPR in terms of scalability and the types of bugs it can effectively address?

---

### Official Review · Reviewer_DEVp · 2024-11-03

**Soundness:** 2
**Presentation:** 2
**Contribution:** 2
**Rating:** 5
**Confidence:** 4

**Summary:**

In this paper the authors have built a tool titled “AutoPR” leveraging LLMs. The authors claim that AutoPR has the ability to automate many of the software development life cycle tasks. They claim that because they leverage LLMs with call-graphs and memory caching they were able to demonstrate zero shot code generation even on new software engineering projects. The authors evaluated AutoPR on the SWE-bench dataset to find its correctness rate is 65.7% better than Devin with only 53.2% (rq1). They argue that the additional information provided by the call-graph has been useful for AutoPR’s performance (rq2). Lastly they also discuss real life challenges with AutoPR or such tools in general (rq3). The authors claim that the novelty in this work is the appropriate combination of routing algorithms, in-memory caching and collaborative technologies that makes “AutoPR” stand out when compared to traditional tools in this space, especially along the lines of efficiency in handling large-scale code analysis.

**Strengths:**

1. Much of the software development tasks are repetitive; therefore this paper attempts to solve a useful problem to save budget, time (effort) that is important from small companies to large enterprises.
2. This paper is largely easy to read and follow.
3. The graph based approaches aiding zero shot code generation seems promising.

**Weaknesses:**

1. Evaluation issues
1a. None of the results are statistically significant p-value? All the research question results should have statistical significance testing applied, and the type of statistical test could be equivalent to the Scott-Knott Test. Using the Scott-Knott Test you could rank different treatments (eg., Devin, A-PR-avg). For a concrete example refer to this publication https://arxiv.org/pdf/1710.09055 section 3.3 here for the use of the Scott-Knott Test.
1b. Not enough repeats to mitigate randomness of the generated patches. It is important have sufficient repeats (say > 20) to analyze the distribution of treatment scores % and use the same in the Scott-Knott test above.
2. This paper reads like a tool paper rather than of research value to be suited for the main track of ICLR. The RQs evaluate a tool rather than a general problem in this space. Please discuss a central research question that literature reported but failed to answer. Then answer that question empirically such that many tools similar or better than AutoPR could be built using insights of the answer to the central research question. On similar lines, what is the future of AutoPR?
3. Related work is at a very high-level discussing program repair and LLMs. Are there no tools similar or remotely similar to what AutoPR does (with or without LLMs) ? If there are no tools even remotely similar, then please share the results of literature survey to validate the same. If there are tools similar to AutoPR please discuss what is the novelty in AutoPR when compared to the others.

4. This paper hints LLM influence here is why,
4a. Weird title or grammar issue
4b. The maths in section 2.1 to 2.3 seems retrofitted to the paper not very coherent to the narrative. Please narrate a concrete example (use case) to clarify sections 2.1 to 2.3  in this work.


Minor issues:

1. Consider rephrasing the last 7 lines in the abstract as they feel disjoint from the nice abstract of the first few lines.
2. Double Quotes `` ‘’ issue in Section in Introduction in the words, New Feature, displays
3. (for brevity we use AutoPR to denote AutoPR…)

**Questions:**

1. Why complicate the problem? “To further complicate the problem…” in Section 2.1?
2. How do the advanced algorithms such as memory caching, routing algorithms fit into the Section 2 call-graph methodology?
3. If it takes multiple attempts (time consuming) to generate the right patch then it can be counter productive for the user for simpler tasks. Any metric to capture the same?

---

### Official Review · Reviewer_oKJU · 2024-11-04

**Soundness:** 2
**Presentation:** 2
**Contribution:** 2
**Rating:** 3
**Confidence:** 3

**Summary:**

The paper presents AutoPR, a tool that leverages large language models (LLMs) in conjunction with *routing algorithms*, *in-memory caching*, and collaborative agent technologies to automate complex software engineering tasks, such as program repair, refactoring, and code optimization. AutoPR offers an innovative approach by starting with a static call graph and optimizing it to capture dependencies that help navigate large-scale codebases, and localize the buggy locations for patch generation. Using SWE-Bench as a testbed, the evaluations demonstrate the potential for productivity gains, showing competitive performance with reported baselines, i.e., SWE-Agent and Devin.

**Strengths:**

- The optimized dependence graph captures relevant dependencies while reducing complexity, making it a task-focused representation.
- The incorporation of memory caching and routing algorithms is a practical addition, useful for handling the scale of modern code repositories.

**Weaknesses:**

- *Lack of comparison with a baseline using static call graph*: Evaluating AutoPR against a non-optimized, static call graph would offer clearer insights into the benefits of the proposed optimization process.
- *Missing comparison with AutoCodeRover [1]*: An open agentic workflow, which performs significantly well on the SWE-bench leaderboard. This is useful to compare different bug localization techniques.
- The paper does not provide a detailed explanation of what types of dependencies the optimization process captures and how these dependencies are justified in a software repository’s context.
- The dependency optimization assumes that the initial static call graph can capture all relevant relationships. However, not all dependencies relevant to buggy locations are captured through calls; dependencies can also arise through shared state, event-driven mechanisms, or configuration settings. These implicit dependencies may require analysis beyond the call graph.
- AutoPR’s results are reported as averages over multiple runs (A-pr-avg and A-pr-all), while baseline tools are only evaluated on a single run. This inconsistency could make AutoPR’s metrics, especially A-pr-all, appear artificially improved in comparison.


Minor:
- References to Figures 6, 7a, 9, etc., when the paper has only two figures. In fact, Section 1 refers to Figure 2, but is not consistent with actual figure.
- Section 2.2, both Step 1 and Step 2 have possibly wrongly formatted paragraphs.

**Questions:**

1. Can you explain in more detail the types of dependencies captured by the optimized graph? Specifically, how does the optimization process handle non-call dependencies, such as shared state, event-driven interactions, or configuration-based dependencies?
2. Could you provide more specifics on how the routing algorithms and memory caching are implemented? For example, how does routing optimize traversal through the dependency graph, and how does caching impact performance in practice?
3. The paper demonstrates efficiency gains through the optimized call graph, but can you discuss how this approach scales with increasingly complex or interdependent codebases? Are there limitations in terms of the types or sizes of codebases where AutoPR’s efficiency might decrease?

---

### Note · Authors · 2024-11-19

I have read and agree with the venue's withdrawal policy on behalf of myself and my co-authors.